# Sustainable Production Management Model for Small and Medium Enterprises in Some South-Central EU Countries

Denis Jelačić [1,*], Andreja Pirc Barčić [1,*], Leon Oblak [2], Darko Motik [1], Petra Grošelj [3] and Matej Jošt [2]

1   Department for Production Management, Faculty of Forestry and Wood Technology, University of Zagreb, 10000 Zagreb, Croatia; dmotik@sumfak.unizg.hr
2   Department of Wood Science and Technology, Biotechnical Faculty, University of Ljubljana, 1000 Ljubljana, Slovenia; leon.oblak@bf.uni-lj.si (L.O.); matej.jost@bf.uni-lj.si (M.J.)
3   Department of Forestry and Renewable Forest Resources, Biotechnical Faculty, University of Ljubljana, 1000 Ljubljana, Slovenia; petra.groselj@bf.uni-lj.si
*   Correspondence: djelacic@sumfak.unizg.hr (D.J.); apirc@sumfak.unizg.hr (A.P.B.); Tel.: +385-1-235-24-83 (D.J.); +385-1-235-25-67 (A.P.B.)

**Abstract:** Small and medium enterprises (SMEs) are main generators of employment and economic development in European Union. In Croatia and Slovenia, SMEs in wood processing (C16) and furniture manufacturing (C31) also play an important role in social cohesion and development of rural areas. The aim of this research was to investigate the current situation in SMEs in wood processing and furniture manufacturing regarding driving parameters of business and production management system in the time of a disturbed market situation caused by the COVID-19 global pandemic. Two different questionnaires in an e-mail survey were used to collect data for this research from companies and from experts in the field of management, production and marketing. Various statistical tests were used using seven driving parameters on data collected from 212 SMEs. Also, the Analytic Hierarchy Process (AHP) was used on the data collected from 20 experts. Results showed that companies in the time of pandemic crisis and during the time of major disturbances in supply chain pay the most attention to their production program and quality of their products, and then to marketing activities and situation on the market. According to the results presented in this research, the management model can help SMEs managers, micro and small enterprises in particular, to improve their decision-making process, make the necessary innovations easier and faster, and maintain the sustainable business and production management system of their companies.

**Keywords:** production management model; small and medium enterprises; wood processing and furniture manufacturing; management parameters; analytic hierarchy process

## 1. Introduction

Small and medium enterprises (SMEs) play a key role in the economy of the European Union as the main generators of employment and economic development [1]. They represent a significant part of the economy and industrial system of every country [2,3]. There were more than 25 million SMEs in EU-28 in 2018, which accounted for 99.8% of all enterprises in non-financial business sector [4]. They generated 56.4% of value added, and two out of three jobs were generated by SMEs [5]. Also, SMEs play an important role in confronting sustainability issues such as green production, products and services, increasing consumption of natural resources and social cohesion. [4,5]

According to the Small Business Act of Europe (SBA) Fact Sheets and SMEs basic figures created by European Commission in 2019 for the year 2018, two South-Central EU-28 countries, Croatia and Slovenia [6,7], show results similar to the EU average. In Croatia, 149.541 SMEs represent 99.7% of all enterprises (EU-28 average is 99.8%). They employ 68.9% of all persons employed (EU-28 average is 66.6%) and generate 59.4% of value added (EU-28 average is 56.4%) [6]. In Slovenia, 145.996 SMEs represent 99.8% of

all enterprises (the same as EU-28 average). Slovenian SMEs employ 72.0% of all persons employed (more than EU-28 average of 66.6%) and they generate 64.5% of value added (compared to EU-28 average of 56.4%) [7]. Overall, in the EU-28 there were 58 SMEs per 1000 inhabitants. Croatia is one of the seven Members State where there are fewer than 50 SMEs per 1000 habitants, and Slovenia is among eight Member States where there were more than 80 SMEs per 1000 habitants. In 2018, Croatian SMEs contributed 58% of the total income of the non-financial business sector, and the share of SMEs in export was 53% [8].

In Croatia and Slovenia both, most of the SMEs, especially those in wood processing (C16—Manufacture of wood and of products of wood and cork, except furniture, manufacture of articles of straw and plaiting materials) and furniture manufacturing (C31—Manufacture of furniture) are situated in rural areas. Therefore, they play important role in social cohesion and development of rural areas in every country. Wood-processing and furniture manufacturing companies in Croatia and Slovenia are highly export oriented, and the share of export of C16 is around 5% of all export from the non-financial business sector, while the share of export of C31 is around 4.5%. The percentage of SMEs' exports meet the above numbers that represent total Croatian industry [9].

According to the European Commission Annual Report on European SMEs 2018/2019, wood processing (C16) and furniture manufacturing (C31) are among the non-financial business sectors in which SMEs' share in value added for the sector is between 60% and 80% [4]. Although SMEs make the large majority of all enterprises in C16 and C31, research and development projects in the past were mostly connected with large companies [10,11]. This fact especially applied for wood-processing companies, because large companies have the equipment, personnel and financial assets for providing the necessities and for implementing such research. In contrast, to be able to survive on the market, small and medium enterprises have to be innovative in all possible aspects [12].

In general, along with the strategic management, SMEs should increase their capabilities of information technology for sustainable development [13,14]. Although the knowledge of sustainable practices in SMEs is scarce, good practices should be spread around the SMEs and maintained [15].

Sustainability and development of SMEs is mostly based on innovations [16], but other aspects of business and production management should also be taken care of. Important roles within the small and medium enterprises should be given to management information system [17], marketing, research and development [18], human resources [19], work safety [20], financial issues and controlling [21] and environmental issues [22] as well.

The core business of wood processing companies (C16) and most of the furniture manufacturing and other wood products (C31) is directly connected to a non-polluting, conserving energy, renewable natural resource material: wood. According to CORINAIR, when wood is used as a material in a production process and when it is used as a fuel, the emission of $CO_2$ to the air equals 0, because of the cycle of oxygen and $CO_2$ in living forests. That is why only $NO_x$ and wood dust are considered as polluters when talking about wood processing and furniture manufacturing. Also, in furniture manufacturing there are few environmental issues that need to be considered in the production process daily, like particle boards with 0 emission of formaldehyde, or water based furnishes. That is why wood processing and furniture manufacturing are considered among the most environmentally friendly industries.

Any kind of crisis could have a large impact on small and medium enterprises because of their size and structure. This is especially true if that crisis causes major problems within the supply chain, such as the COVID-19 pandemic causing a period of lockdowns on different levels for different countries. The stability of the supply chain is of great importance for SMEs, and micro and small enterprises are getting more vulnerable to large disturbances. The COVID-19 pandemic and resulting lockdowns caused large disturbances in the market, and is one of the reasons pushing a large number of SMEs into bankruptcy. The rate of business failures among SMEs could be doubled, as some estimations show [23]. Therefore,

small and medium enterprises have to respond to the challenges that the pandemic created on the market and modify their business models [24–27].

Forest based industries, or wood-based industries, are among important manufacturing sectors for European Union economy. Traditional macro-economic indicators, such as GDP growth, increase of industrial production and increase of exports have a positive impact on SMEs [28]. Along with the positive influence by macro-economic parameters, SME managers have to find internal strength by using appropriate methods of financial controlling and managing business and production parameters to improve their performance on the world market [29]. Whatever needed to be done, increased or improved financially or technologically, it is all driven by and related to human resources, which are of great importance in small and medium enterprises. Therefore, corporate culture and human resource management have to be involved in the process of growth and development of SMEs [30].

Regarding the previously mentioned, however important any of the parameters are for improving the performance of any small or medium enterprise, none of them should be considered separately [31]. According to the economic environment, social environment and situation on the market, a specific business and production management model should be established to help SMEs to survive or improve.

Therefore, as the first aim of this research, we investigate the current situation in small and medium enterprises in wood processing (C16) and furniture manufacturing (C31), focusing on driving parameters of business and production management systems in the time of the disturbed market situation caused by the COVID-19 global pandemic. According to the given situation, the second aim of this research was to create an applicable and sustainable business and production management model for SMEs in C16 and C31 in two south-central EU countries (Croatia and Slovenia).

## 2. Materials and Methods

In order to investigate the importance of the seven driving parameters of the business and production management system, two types of data were collected. Seven driving parameters established in the previous research were derived from 29 individual criteria, placed into seven groups according to business or production management departments they belong to [31]. The first type of data was collected by interviewing owners and top managers of small and medium enterprises in wood processing (C16) and furniture manufacturing (C31). The second type of data was collected from experts who pairwise compared the driving parameters using the Analytic Hierarchy Process (AHP).

### 2.1. Survey

A survey was sent to the company managers of micro/small and medium enterprises in C16 and C31 in Croatia and Slovenia. The questionnaire in the survey consisted of five questions on company characteristics related to manufacturing program, size of the company, type of production and technology in the company, and management model. The main part of the questionnaire was a ranking of seven driving parameters of the business and production management system. The survey was conducted in late 2020 and early 2021. The target population of the survey were SMEs in wood processing (C16) and furniture manufacturing (C31).

In Croatia, the sampling frame were enterprises in C16 and C31 which email addresses were in public access databases. An emailed survey, based on methods recommended by Dillman [32], was the approach used in this study. Questionnaires were sent by email to 246 companies. A total of 187 companies confirmed acceptance of the email, while the others no longer exist, have changed their email addresses or are not directly related to wood processing and furniture manufacturing. The total of 81 responses were received. The response rate was 43.3%.

In Slovenia, the survey was conducted via 1KA, an open-source online survey application. The sampling frame were enterprises in C16 and C31 with more than 5 employees,

which email addresses are in public access databases. The survey link was sent to 621 companies and 131 completed the survey. The response rate was 21.1%. The response rate in both countries was sufficient for analysis purposes in business and enterprise surveys [33].

Data were analyzed in SPSS Statistics 25.0. The level of statistical significance was set at $\alpha = 0.05$. The non-parametric Friedman test was used to test the difference between seven driving parameters, with post hoc Wilcoxon signed-rank tests to examine the differences between 6 pairs of sequentially ranked driving parameters. A Bonferroni adjusted significance level (0.05/6 = 0.0083) was used. Parameters given in the survey were:

- LPOSC–Leadership, Policy, and Organizational Structure of the Company
- PCMPPD–Process Culture, Management Processes, and Production Deadlines
- RPQP–Range of Products and Quality of Products
- MMAC–Marketing and Market Activities of the Company
- HR–Human Resources
- ITMPT–Information Technology and Modern Production Technology
- EFP–Environmentally friendly production.

When $n$ objects are ranked, the agreement and correlation among $m$ sets of the ranking can be measured. Spearman's rank-order coefficient of correlation was used for two sets of ranking and the Kendall's coefficient of concordance W [34] for more than two sets of ranking. Both coefficients can take values on the interval [0, 1], with no agreement at value 0 and perfect agreement among sets of ranks at value 1.

### 2.2. Analytic Hierarchy Process (AHP)

Analytic Hierarchy Process (AHP) [35] is a method that allows comparing and ranking the importance of multiple objects (criteria, alternatives, etc.). It can combine quantitative and qualitative analysis. It is one of the widely used multi-criteria decision-making methods that has been applied recently to solve many complex decision problems. Oblak et al. [36] used AHP to evaluate the factors in buying decision process of furniture consumers. Camci et al. [37] used hesitant fuzzy AHP to select the best computer numerical control router for woodwork manufacturing based on 4 criteria and 11 subcriteria. Goyal et al. [38] used fuzzy AHP to prioritize the barriers in achieving sustainable production and consumption in the manufacturing industry. To assess the challenges to sustainable humanitarian supply chain management, Karuppiah et al. [39] used neutrosophic AHP. The relative importance of objects at the same hierarchy level is expressed by pairwise comparisons of objects. The fundamental Saaty's ratio scale of 1 to 9 is used for the judgments, where 1 expresses that the compared objects are equally important and 9 that one object is absolutely more important than the other. The judgments are collected in a pairwise comparison matrix $A$:

$$A = \left(a_{ij}\right)_{n \times n} = \begin{bmatrix} a_{11} & a_{12} & \cdots & a_{1n} \\ a_{21} & a_{22} & \cdots & a_{2n} \\ \vdots & \vdots & \ddots & \vdots \\ a_{n1} & a_{n2} & \cdots & a_{nn} \end{bmatrix}, \ a_{ii} = 1, \ a_{ji} = 1/a_{ij} \tag{1}$$

The vector of weights $w = (w_1, w_2, \ldots, w_n)$ can be derived from $A$ by eigenvector method:

$$Aw = \lambda_{\max} w \tag{2}$$

where $\lambda_{\max}$ is the principal eigenvalue of $A$. Consistency of judgment is an important issue in AHP, measured by consistency ratio CR, the ratio of the consistency index (CI) and the random index (RI) [40]:

$$CR = CI/RI, \ CI = (\lambda_{\max} - n)/(n - 1) \tag{3}$$

Saaty gave the acceptable consistency as CR < 0.1, but in this research CR < 0.15 was considered acceptable [41], which helped the experts in the evaluation. The weighted geo-

metric mean method, most commonly used in applications [42,43], was used to aggregate individual judgments into group judgments. A total of 20 wood processing and furniture manufacturing experts in management, production and marketing were selected, 10 from Croatia and 10 from Slovenia. The Microsoft Excel software template [44] was used for evaluation, which also calculated the level of consensus.

*2.3. Simple Multi-Attribute Rating Technique Exploiting Ranks (SMARTER)*

To compare the results of the survey and the AHP, the ranks of the driving parameters $r_1, r_2, \ldots, r_n$ must be converted into importance weights $w_1, w_2, \ldots, w_n$ However, the weights can be strongly influenced by the computation method. The method should produce intuitive and accurate weights. Various methods have been proposed in literature to determine the weights based on the ranks. The rank order weighting method SMARTER, also called the rank-order centroid weight method (ROC) [45]. SMARTER was chosen as a method with good theoretical justification that performs better than other similar methods in terms of selection accuracy [46] and robustness [47]:

$$w_j = \frac{1}{n} \sum_{k=j}^{n} \frac{1}{r_k} \qquad (4)$$

It has been used to rank long-term forest management scenarios [48], to rank sustainability assessment criteria for large-scale composting technologies [49], and to rank environmental criteria in the life cycle assessment interpretation method [50], among others.

To combine survey and AHP results, the final weights were calculated as their linear combination:

$$w_{final} = \alpha w_{SMARTER} + (1 - \alpha) w_{AHP} \qquad (5)$$

where $\alpha \in [0, 1]$ is a factor of the importance of the survey results.

**3. Results**

For the purposes of the investigation of the current situation and perception of the driving parameters of business and production management among the small and medium enterprises in in wood processing (C16) and furniture manufacturing (C31) in Croatia and Slovenia, a total of 867 questionnaires were sent to enterprises, of which 246 were in Croatia and 621 in Slovenia. A total of 212 questionnaires were completed, of which 81 (38.2%) were in Croatia and 131 (61.8%) in Slovenia, with a response rate of 43.3% in Croatia and 21.1% in Slovenia.

The first part of the questionnaire sent to companies consisted of questions giving the profile of the companies participated in the survey. In this part of the questionnaire were the questions about the production program (16 groups of products from the wooden boards to final products such as furniture or joinery), the size of the enterprise, type of the production process, production management model and type of the technology used in the production process. Results are presented in the Table 1.

**Table 1.** Company profile: size of the enterprise; production process type; type of technology used in the production process; production management model.

| | Croatia | Slovenia | All | | Croatia | Slovenia | aAll |
|---|---|---|---|---|---|---|---|
| Response share | 38.2% | 61.8% | 100.0% | Mostly computer aided technology | 23.5% | 11.5% | 16.0% |
| Less than 10 employees (micro) | 29.6% | 59.5% | 48.1% | Mostly classic technology | 22.2% | 19.1% | 20.3% |
| Less than 50 employees (small) | 40.7% | 32.1% | 35.4% | Mostly hand tools and machines | 9.9% | 11.5% | 10.8% |
| Less than 250 employees (medium) | 29.6% | 8.4% | 16.5% | A combination of all of the above | 44.4% | 58.0% | 52.8% |
| Individual production | 35.8% | 61.8% | 51.9% | Work for a known customer | 63.0% | 74.8% | 70.3% |
| Small series production | 39.5% | 27.5% | 32.1% | Work for an unknown customer (showrooms and retail stores) | 0.0% | 1.5% | 0.9% |
| Serial production | 24.7% | 10.7% | 16.0% | A combination of work for a known and an unknown customer | 37.0% | 23.7% | 28.8% |

### 3.1. Results of the Ranking of Business and Production Management Parameters in the Survey

The second part of the questionnaire sent to companies consisted of the seven driving parameters of business and production management, which company owners or top managers should have ranked according to their perception of the importance of each parameter.

The results of the survey are presented as average ranks of the driving parameters in Table 2.

**Table 2.** Ranking of driving parameters of business and production management system by average ranks from survey, SMARTER weights, AHP weights and final weights for $\alpha = 0.5$.

|  | LPOSC | PCMPPD | RPQP | MMAC | HR | ITMPT | EFP |
|---|---|---|---|---|---|---|---|
| Mean rank | 3.7 | 3.4 | 2.3 | 4.7 | 4.0 | 4.5 | 5.5 |
| Ranks | 3 | 2 | 1 | 6 | 4 | 5 | 7 |
| SMARTER weights | 0.162 | 0.161 | 0.255 | 0.106 | 0.134 | 0.110 | 0.072 |
| AHP weights | 0.137 | 0.111 | 0.170 | 0.232 | 0.126 | 0.160 | 0.063 |
| Ranks | 4 | 6 | 2 | 1 | 5 | 3 | 7 |
| Final weights | 0.150 | 0.136 | 0.213 | 0.169 | 0.130 | 0.135 | 0.068 |
| Ranks | 3 | 4 | 1 | 2 | 6 | 5 | 7 |

LPOSC—Leadership, Policy, and Organizational Structure of the Company; PCMPPD—Process Culture, Management Processes, and Production Deadlines; RPQP—Range of Products and Quality of Products; MMAC—Marketing and Market Activities of the Company; HR—Human Resources; ITMPT—Information Technology and Modern Production Technology; EFP—Environmentally friendly production.

The Friedman test confirmed statistically significant differences between the mean ranks of the driving parameters ($\chi^2(6) = 299$, $p < 0.001$). Companies ranked RPQP as by far the most important driving parameter with the mean rank of 2.28. Post hoc analysis using Wilcoxon signed rank tests showed a significant difference between RPQP and PCMPPD (3.36) in second place ($z = -6.845$, $p < 0.001$). LPOSC (3.71) ranks third with non-significant difference from PCMPPD ($z = -1.630$, $p = 0.062$) and from HR (3.98) ranked forth ($z = -1.258$, $p = 0.212$). ITMPT (4.51) ranked fifth with significant difference from HR ($z = -2.926$, $p = 0.003$) and non-significant difference from MMAC (4.70), ranked sixth ($z = -1.224$, $p = 0.221$). EFP was clearly the least important driving parameter with mean rank 5.45, with significant difference from MMAC ($z = -3.969$, $p < 0.001$).

Kendall's coefficient of concordance was calculated to determine the degree of agreement among companies. The concordance among companies is quite low ($W = 0.235$, $p < 0.001$), indicating that the companies have diverse opinions about which driving parameters are important.

Spearman's rank-order coefficient confirms a significant strong positive correlation ($r_S(7) = 0.883$, $p = 0.008$) between the mean ranks of Croatian and Slovenian companies. Both Croatian and Slovenian companies evaluated RPQP as the most important driving parameter (Figure 1). However, its average rank was lower in Slovenia (1.98) than in Croatia (2.75). In Croatia, HR was the second most important driving parameter, while PCMPPD was second in Slovenia. Both countries assessed RPQP, LPPOSC, PCMPPD and HR as the four most important driving parameters.

Spearman's rank-order coefficient shows a fairly strong positive, though not significant, correlation ($r_S(7) = 0.714$, $p = 0.071$) between the mean ranks of micro and small companies and between small and medium companies ($r_S(7) = 0.679$, $p = 0.094$). The correlation between micro and medium companies is less strong ($r_S(7) = 0.464$, $p = 0.294$). Micro and small companies evaluated RPQP as the most important driving parameter (Figure 2) and HR as the second most important. On the other hand, medium companies highlighted LPOSC as the most important driving parameter, followed by PCMPPD and RPQP.

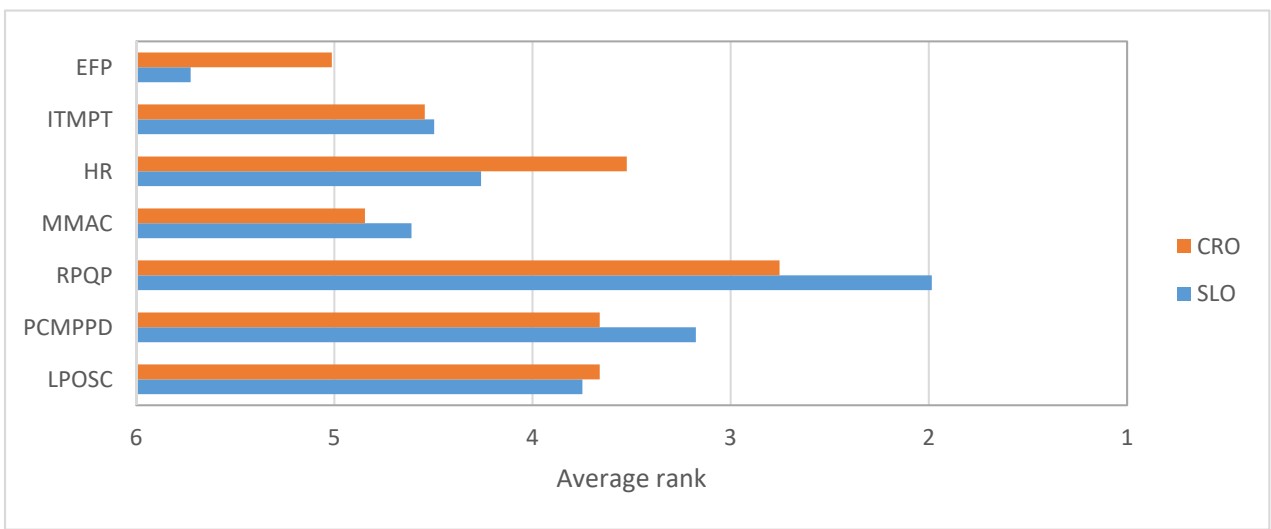

**Figure 1.** Average ranks for Croatia and Slovenia.

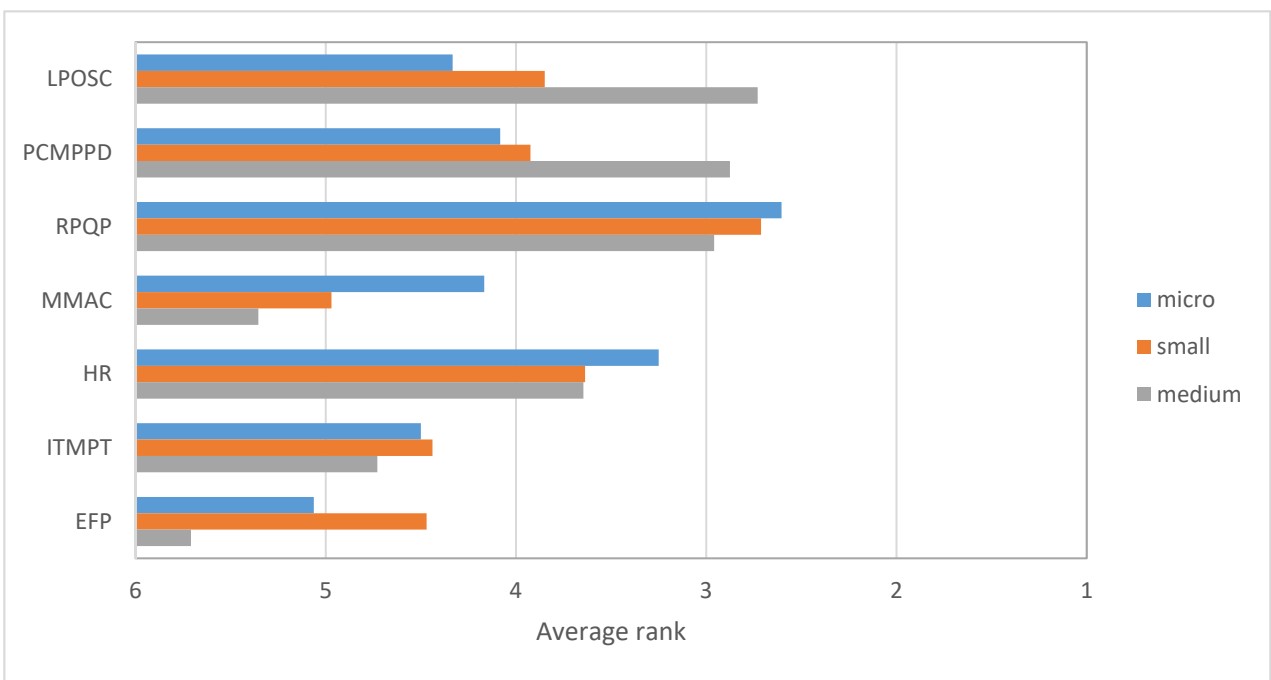

**Figure 2.** Average ranks regarding the size of the companies.

### 3.2. AHP Results

A total of 20 wood processing and furniture manufacturing experts in management, production and marketing, 10 from Croatia and 10 from Slovenia, were chosen to give their judgements in a different questionnaire, prepared for the AHP analysis, and to derive the importance of each driving parameter through pairwise comparison. Their judgments were aggregated by geometric mean into group judgments (Figure 3).

The AHP weights are presented in Table 2 and Figure 3. The consensus level was 70.5%, indicating a moderate level of agreement among experts. The results show that experts assessed MMAC as the most important driving parameter, followed by RPQP and ITMPT.

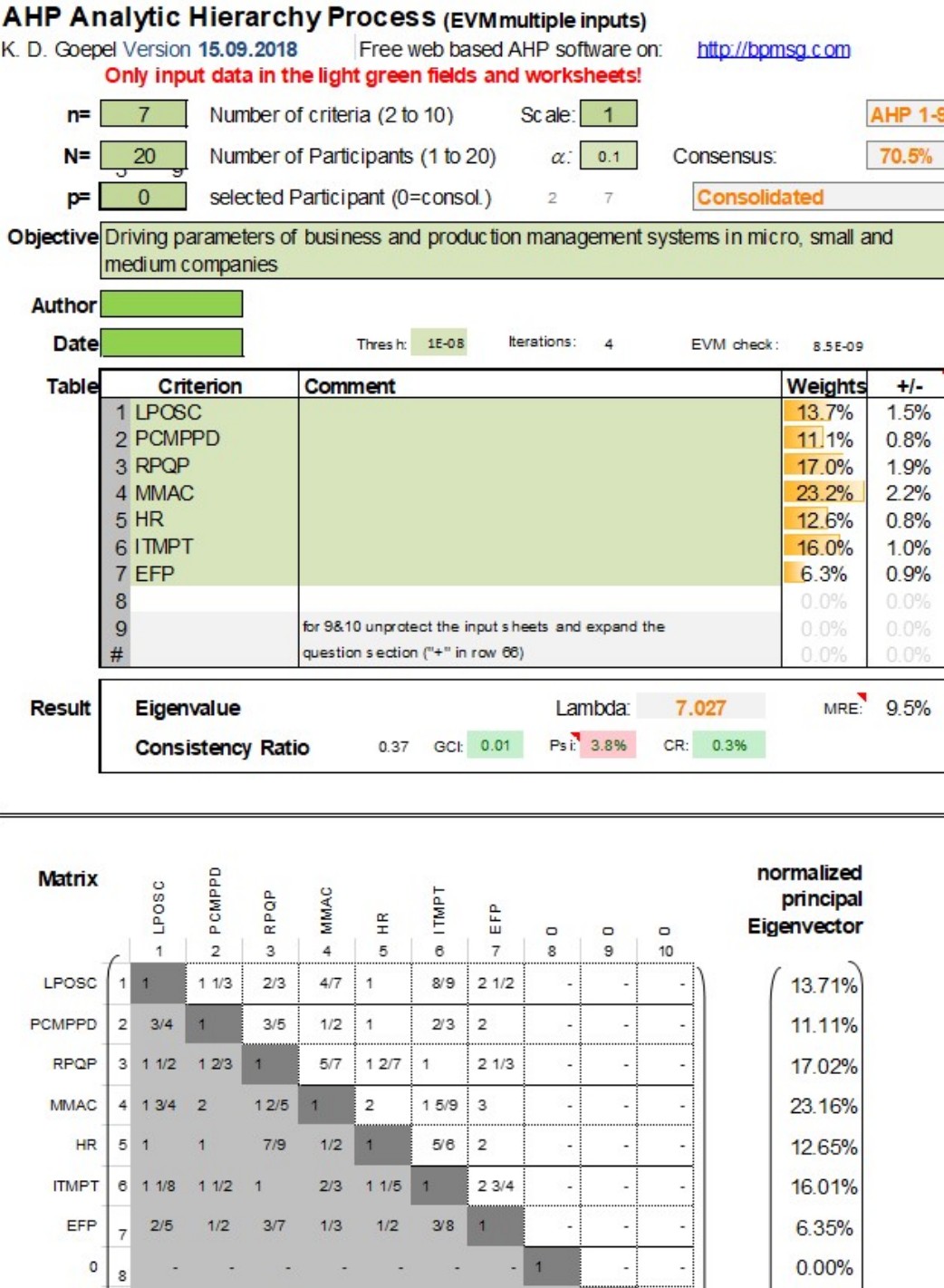

**Figure 3.** AHP Analysis on the Business and Production Management Parameters.

Both Croatian and Slovenian experts evaluated MMAC as the most important driving parameter (Figure 4). However, while Croatian experts evaluated LPOSC as the second most important driving parameter, Slovenian experts put ITMPT on the second place. RPQP and HR were ranked third and fourth by Croatian and Slovenian experts.

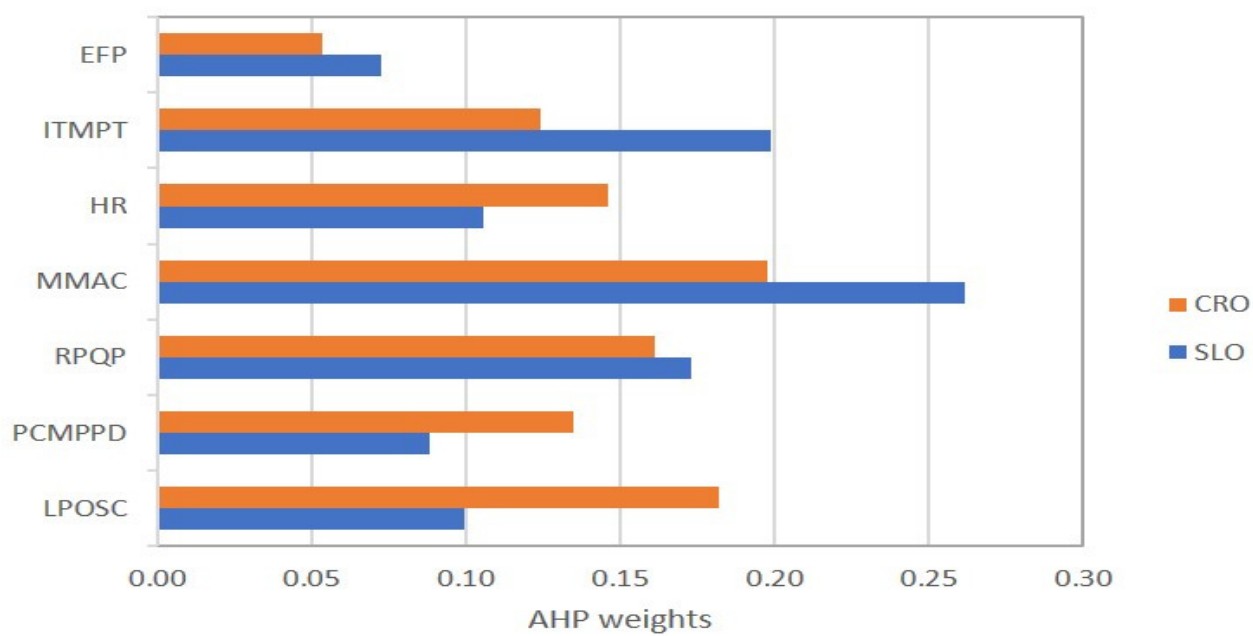

**Figure 4.** AHP weights for Croatia and Slovenia.

The rankings of driving parameters between experts (AHP) and companies (survey) differ significantly. Figure 5 shows that there is a very poor correlation between survey and AHP rankings ($r_S(7) = 0.143$, $p = 0.760$).

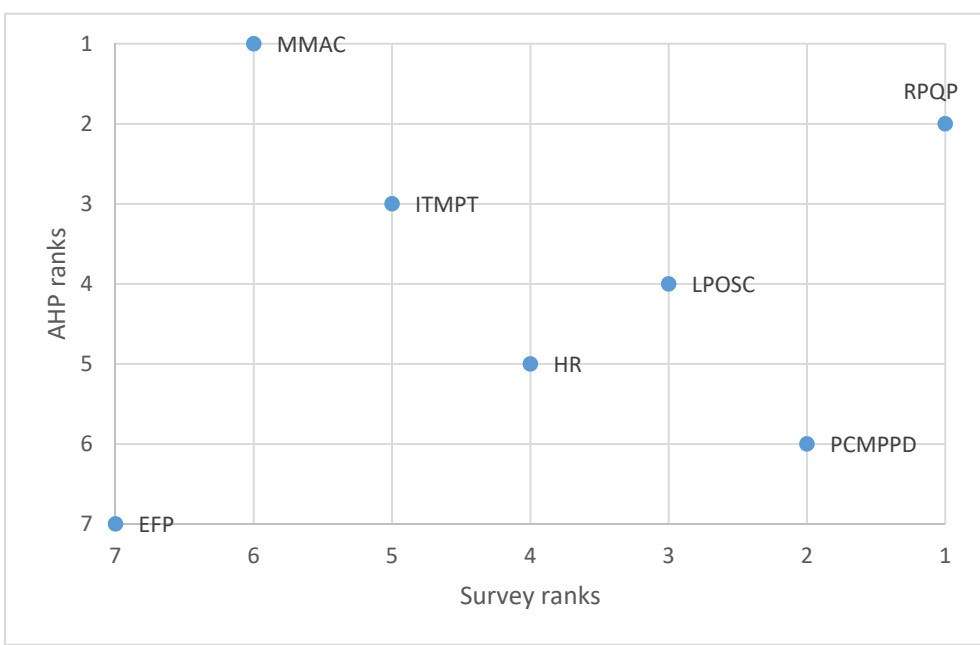

**Figure 5.** Ranking of driving parameters by survey and AHP.

### 3.3. The Final Weights of Driving Parameters

To convert the survey ranking into weights of importance of driving parameters SMARTER (Equation (4)) was used and the results are presented in Table 2. The final weights were calculated as linear combination of survey and AHP weights using Equation (5). Figure 6 presents final weights as a function of factor of importance α.

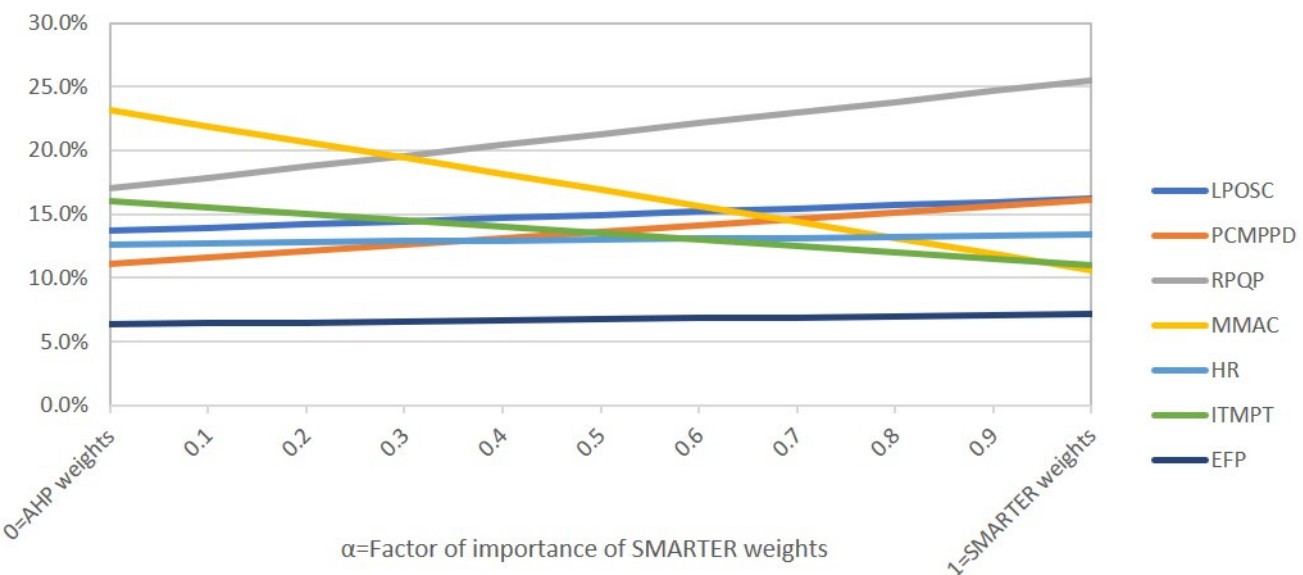

**Figure 6.** Final weights depending on the factor of importance of SMARTER weights.

While companies usually emphasize only one or two driving parameters, experts have a more holistic view that considers all driving parameters. Table 2 shows the final weights and ranks of the driving factors for α = 0.5 when both types of weights are equally important. The rank of the first three most important driving parameters does not change when the factor α takes values from 0.4 to 0.6.

## 4. Discussion

The first aim of this research was to investigate the current situation in small and medium enterprises in wood processing (C16) and furniture manufacturing (C31) regarding driving parameters of the business and production management system in a time of disturbed market situation caused by the COVID-19 global pandemic.

According to the results of the research conducted in Croatia and Slovenia, two south-central EU countries, in which C16 and C31 play an important role of the economy, especially regarding SMEs, the disturbances in the supply chain during the lockdown of EU market changed the way of thinking in production systems and doing business. The average ranks of the driving parameters of the business and production system among SMEs entrepreneurs and company managers in Croatia and Slovenia both consider production program, range of products and quality of the product by far the most important driving parameter in their management systems. Although in Croatia managers rank human resources as the second most important driver, and Slovenian managers rank process culture, management processes and production deadlines as the second most important driver, managers in both countries agree that those three drivers, along with leadership, policy and organizational structure of the company are the four drivers more important than the remaining three.

The results of the AHP analysis of the experts' rankings of the same drivers, showed that experts consider particular driver of management differently than managers in companies. The experts consider marketing and market activities by far the most important management parameter, followed by production program and information technology.

The opinions of entrepreneurs and managers in C16 and C31 SMEs in this research were considered equally important as the opinions of the experts. Therefore, the final weights of the particular business and production management parameter showed their overall ranking.

1. Range of products and quality of products—RPQP
2. Marketing and market activities of the company—MMAC
3. Leadership, policy and organizational structure of the company—LPOSC
4. Process culture, management processes and production deadlines—PCMPPD
5. Information technology and modern production technology—ITMPT
6. Human resources—HR
7. Environmentally friendly production—EFP.

Comparing the results of this research with the results of the research conducted in the year 2017 [31], before the COVID-19 global pandemic crisis, in a different business environment, it can be confirmed that the way of thinking about the management drivers in business and production systems has changed, among managers and experts both. Previous research [31] was conducted in four south-east European countries, including Croatia and Slovenia. Four years ago, the ranking was as follows:

1. Marketing and market activities of the company—MMAC
2. Range of products and quality of products—RPQP
3. Information technology and modern production technology—ITMPT
4. Process culture, management processes and production deadlines—PCMPPD
5. Human resources—HR
6. Leadership, policy and organizational structure of the company—LPOSC
7. Environmentally friendly production—EFP.

According to the results of this research, it can be observed that entrepreneurs and managers in SMEs have turned more to assets they have and to those which are reachable including quality of their products and changes in the production program, innovations in leadership and organizational structure of the company instead of modern production and information technology or human resources, mostly because of the disturbances in supply chain caused by lockdowns, but because of the other market conditions caused by the pandemic crisis as well.

Differences in opinion can be observed between micro and small enterprises on one side, and medium enterprises on the other. These differences are understandable starting from their size in the first place. Medium size companies have more personnel and better financial assets to deal with many different aspects of management. Therefore, it is understandable that micro and small companies consider production program and quality of their products as the most important driving parameter, followed by human resources as the second most important, while medium enterprises consider leadership, policy and organizational structure as the most important driver, followed by process culture and management processes. Medium enterprises ranked production program as the third most important management parameter.

Although C16 and C31 are industries highly sensible to environmental issues, dealing with wood which is a renewable natural resource, respondents ranked environmentally friendly production as the last one. In the moment of crisis, especially in small and micro companies, when the supply chain is heavily disturbed, it is only logical that managers and owners think about survival first (economic sustainability), and how to keep their employees (economic and social sustainability) and environmental issues later. However, it doesn't mean that environmental issues should be left out of any discussion on sustainability. After all, environmental issues are at the core of the business of wood processing and furniture manufacturing.

The COVID-19 pandemic crisis changed the society in many different ways. It also changed the entrepreneurship and management system in general [51]. Different authors were investigating the possibilities of creating business models in different production and service sectors by exploring transformation drivers for changes in organization, technology or finances [24]. Some of them saw the pandemic crisis as a major opportunity for SMEs to change their business and production model, to invest in research and development more than usual [52]. Others recommended to small and medium enterprises to invest more into digitalization and information technology [53], since during the COVID-19 global

pandemic most of the financial activities, marketing and trade activities and business arrangement went online [54].

According to the given situation with the COVID-19 global pandemic, and according to the results achieved with this research, the second aim was to create an applicable business and production management model for SMEs in C16 and C31 in Croatia and Slovenia. The aim was to create an organization model for small and medium enterprises which can help make decision process in a company easier, faster and which could meet the requirements of the turbulent and ever-changing market for wood products and furniture.

The newly established model was created based on the final weights ranking (Table 2) and it meets the requirements placed by the seven production and business management parameters. Usually, the organization models place different functions or group of parameters of the same of similar importance on the same level. The parameters in this model are almost at the same level and they are almost equally important in decision making process, but some of the parameters or groups of parameters, according to the results of this research, have slightly higher priority. Therefore, they are not at exactly the same level, but slightly moved up or down according to priority they achieved in the analysis. A similar model was created in the year 2018, in the normal business and production environment, before the major market disturbances caused by the COVID-19 pandemic crisis [55].

The model (Figure 7) is flexible and if the company management makes a decision to go into the innovations in information technology or in human resources, it could bring particular group of parameters up front in the model and make the decision-making process easier, faster and more effective.

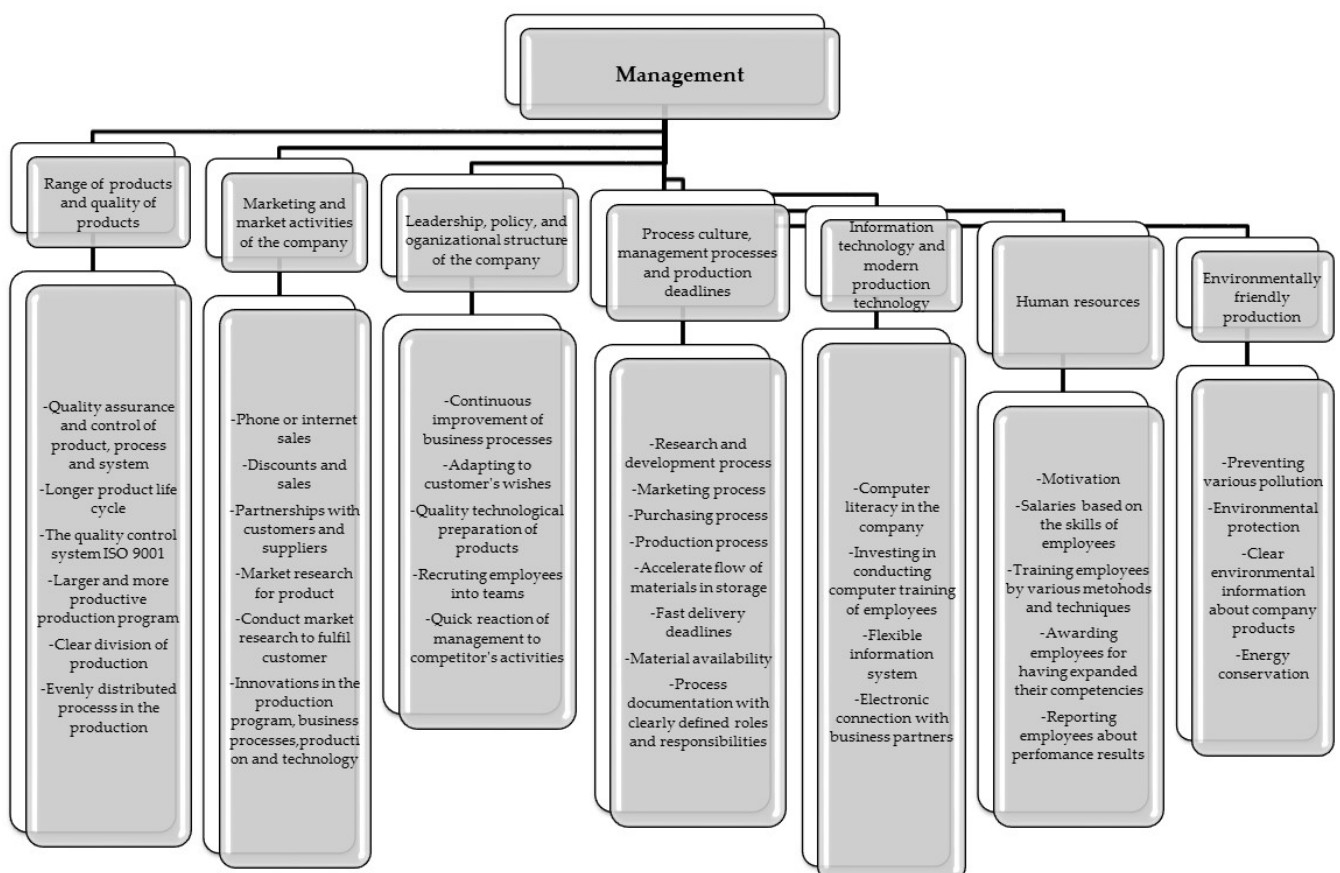

**Figure 7.** Organization management model for small and medium enterprises.

Also, small and medium companies usually do not have enough personnel to deal with all those issues at the same time and discuss them during meetings or otherwise. That

is especially relevant to small and micro companies, where one or two persons have to make decisions on all issues. Therefore, they need some kind of model which will give priorities to some issues over the other and make the whole decision-making process easier and faster.

Although the model was created according to the results of this research during the COVID-19 global pandemic crisis, because of its flexibility, it can be used not only in present pandemic business conditions, but in different business environments as well. Also, the model can help SMEs to avoid business failures, since some estimations given by International Monetary Fund [23] predict for the manufacturing sector to double the number of business failures (8.64% pre-COVID-19 to 16.94% COVID-19). The same research predicts for Slovenia to increase the bankruptcy rate in SMEs from 7.27% (pre-COVID-19) to 17.26% (COVID-19), and the similar could happen in Croatia. The Croatian government is trying to avoid that scenario by subsidizing SMEs in C16 and C31, as one of the most important manufacturing sectors, helping them to survive this crisis. Some authors of this research are in the process of collecting data for the next research trying to find the exact impact of COVID-19 to business failures in SMEs in C16 and C31. Some future research, in a different business environment, in a time with less disturbances on the market and with a stable supply chain, may confirm its usability and efficiency for small and medium enterprises, not only in wood processing (C16) and furniture manufacturing (C31).

## 5. Conclusions

The aim of this research was to establish the differences among the drivers for business and production management systems in small and medium enterprises for wood processing and furniture manufacturing in Croatia and Slovenia during the COVID-19 global pandemic crisis.

The results of the research showed that, in effort to survive the crisis, entrepreneurs and managers in SMEs turned their way of thinking and their managerial skills to inner strength. Since SMEs should be innovative to survive and to develop on the market, managers focused on production program and quality of their products, since the lockdown and pandemic made it much harder to be innovative on the market or to put efforts towards finding new markets.

This research, analysis and provided model can help managers in SMEs in wood-processing and furniture manufacturing improve their decision-making process, to make it sustainable, improving their production and business results.

**Author Contributions:** Conceptualization, D.J., L.O. and D.M.; methodology, D.J., M.J. and P.G.; software, M.J. and P.G.; validation, D.J., L.O. and A.P.B.; formal analysis, P.G.; investigation, D.J., L.O., A.P.B. and D.M.; resources, A.P.B.; data curation, D.J. and M.J.; writing—original draft preparation, D.J.; writing—review and editing, D.J., A.P.B. and L.O.; visualization, D.J.; supervision, D.J. and L.O.; project administration, M.J.; funding acquisition, D.J., D.M., A.P.B., P.G. and L.O. All authors have read and agreed to the published version of the manuscript.

**Funding:** Denis Jelačić: Andreja Pirc Barčić and Darko Motik wish to thank University of Zagreb, Faculty of Forestry and Wood Technology, Fund for scientific and professional work for financial support. Leon Oblak, Petra Grošelj and Matej Jošt acknowledge the financial support from Slovenian Research Agency (P4-0015; P4-0059).

**Institutional Review Board Statement:** Not applicable.

**Informed Consent Statement:** Not applicable.

**Data Availability Statement:** The data presented in this article are available on request from the corresponding author. The data are not publicly available due to privacy restrictions.

**Conflicts of Interest:** The authors declare no conflict of interest. The funders had no role in the design of the study; in the collection, analyses, or interpretation of data; in the writing of the manuscript, or in the decision to publish the results.

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
