# Peer review of "Sustainable Production Management Model for Small and Medium Enterprises in Some South-Central EU Countries"

_sustainability, doi:10.3390/su13116220_

Round 1

Reviewer 1 Report

Although the study was done only for two countries in South-Central Europe, it is relevant for most of the EU28 countries and the rest of the Balkan countries. The article is up to date, well structured and well written. On the basis of the research done and subsequent processing of the results, very interesting analyses, conclusions and comparison with previous research are made.

The only question that remains open is what is the number of companies in Croatia and Slovenia affected by Covid 19 and, in particular, the percentage of companies that went bankrupt. If the authors can draw a conclusion from their research, this will be interesting information.

Reviewer 2 Report

The article is well written, and the topic and results are important for business management, but the aims is not the focus of sustainability.

The definition of Sustainable Production is the creation of goods and services using processes and systems that are: non-polluting, conserving of energy and natural resources, economically viable, safe, and healthful for workers, communities, and consumers, Socially and creatively rewarding for all working people. In this way, I think so that your paper aims about economical viability, but the environmental al social aspect is not addressed.

It is important that they talk a little about the other dimensions of sustainability (social and environmental) since the whole focus is on economic sustainability. I see that the study results in that the last thing to consider is "environmentally friendly production", so you cannot speak of a sustainable production if the environment is not considered.

Please include in your article the other two dimensions of sustainability: social and environmental.

Reviewer 3 Report

This paper is appropriate for the journal and timely. Some minor suggestions include

-providing the significance level at which all hypotheses were tested. I assumed this was 0.05 based on some of the text in the results.

-Please consider providing more background information on the SMARTER and AHP methods. Readers less familiar with these methods would benefit from some practical background information, particularly related to how/when/why to use them.

Round 2

Reviewer 2 Report

It´s Ok. Thank you